# Circulating RNA Markers Associated with Adenoma–Carcinoma Sequence in Colorectal Cancer

**DOI:** 10.3390/ijms26041518

**Published:** 2025-02-11

**Authors:** Li Ah Kim, Jin Han, Tae Il Kim, Jae Jun Park, Jae Myun Lee, Jong Koo Kim, Sunyoung Park, Hyeyoung Lee

**Affiliations:** 1Department of Biomedical Laboratory Science, College of Software and Digital Healthcare Convergence, Yonsei University Mirae Campus, Wonju 26493, Republic of Korea; riah0416@naver.com (L.A.K.); kristenlovemom@gmail.com (J.H.); 2Division of Gastroenterology, Department of Internal Medicine, Yonsei University College of Medicine, Seoul 03722, Republic of Korea; taeilkim@yuhs.ac (T.I.K.); jaejpark@yuhs.ac (J.J.P.); 3Department of Microbiology and Immunology, Institute for Immunology and Immunological Diseases, Yonsei University College of Medicine, Seoul 03722, Republic of Korea; jaemyun@yuhs.ac; 4Department of Family Medicine, Wonju College of Medicine, Yonsei University, Wonju 26426, Republic of Korea; kimjk214@yonsei.ac.kr; 5School of Mechanical Engineering, Yonsei University, Seoul 03722, Republic of Korea; angelsy88@gmail.com; 6INOGENIX Inc., Chuncheon 24232, Republic of Korea

**Keywords:** adenoma–carcinoma sequence, colorectal cancer, differentially expressed genes, RNA, RNA transcripts

## Abstract

Colorectal cancer progresses through a well-defined adenoma–carcinoma sequence (ACS), which is pivotal for early detection and intervention. While ACS-based surveillance has been instrumental, its reliance on tissue sampling limits accurate staging. Liquid biopsies, including circulating tumor DNA (ctDNA) and extracellular RNA, have emerged as non-invasive alternatives, yet they primarily detect genetic alterations or passive RNA release rather than active biological processes. Thus, there is a need for biomarkers that reflect real-time immune responses and tumor–microenvironment interactions during ACS progression. This study aimed to identify circulating RNA biomarkers associated with ACS by analyzing blood samples from 160 individuals across five groups: colorectal cancer, advanced adenoma, non-advanced adenoma, symptomatic non-disease control, and healthy control. RNA sequencing coupled with gene ontology and protein–protein interaction analyses identified stage-specific circulating transcripts. Notably, *IFI27* was linked to the symptomatic non-disease control group, *DEFA4* to the non-advanced adenoma group, *MPO* to the advanced adenoma group, and *CD177* to the colorectal cancer group. These findings suggest that colorectal-cancer-related circulating RNA markers reflect host immune responses during ACS progression, supporting their potential role in early detection and non-invasive diagnoses. By addressing critical gaps in early colorectal cancer detection, this study advances the utility of circulating RNA biomarkers and liquid biopsies in colorectal cancer screening and clinical management.

## 1. Introduction

Colorectal cancer is the third leading cause of cancer-related morbidity and mortality worldwide, with over 1.8 million new cases and approximately 881,000 deaths each year [1]. A key challenge in the management of colorectal cancer is the lack of early symptoms, which leads to a delayed diagnosis. Consequently, approximately 20% of colorectal cancer cases are detected at an advanced stage when metastatic disease is already present. Even among patients diagnosed at an early stage, more than 30% eventually develop metastatic disease, resulting in poor survival outcomes [2,3].

Colorectal cancer typically arises from adenomatous polyps and benign growths in the colon or rectum, which may progress to malignant carcinoma via the adenoma–carcinoma sequence (ACS). This model, first described by Vogelstein et al., outlines the key genetic mutations driving colorectal cancer progression, including early mutations in the *APC* gene, followed by *KRAS* mutations during adenoma progression, and *TP53* mutations frequently observed in invasive carcinomas [4,5]. The early detection and removal of adenomatous polyps via a colonoscopy are essential to prevent colorectal cancer progression [6].

Despite advances in understanding colorectal cancer’s molecular pathogenesis, clinical applications of this knowledge are limited by the need for invasive tissue sampling. Non-invasive methods, such as stool-based DNA tests, have been developed for colorectal cancer screening, but they have limited sensitivity, detecting only 43% of advanced adenomas and 14% of non-advanced adenomas [7,8]. These limitations emphasize the need for more sensitive and non-invasive biomarkers that can detect colorectal cancer at earlier stages and track disease progression.

Liquid biopsies have emerged as a promising alternative for cancer detection, enabling the non-invasive monitoring of tumor-associated biomarkers, including circulating tumor DNA (ctDNA), circulating tumor cells (CTCs), and circulating RNA transcripts [9,10,11]. While ctDNA primarily reflects genetic mutations and CTCs provide insights into metastatic potential, circulating RNA transcripts offer a more dynamic perspective by capturing real-time transcriptional activity within tumor-associated cells [12,13]. Unlike extracellular miRNAs, which are passively released and often reflect systemic changes rather than tumor-specific processes, intracellular circulating RNA transcripts provide a functional snapshot of active gene expression, particularly in immune responses and tumor–microenvironment interactions [14,15]. This highlights their potential as sensitive biomarkers for detecting the adenoma–carcinoma sequence (ACS) and monitoring early colorectal cancer (CRC) progression.

In recent years, circulating RNA transcripts have shown potential as diagnostic and prognostic tools for several cancers, including colorectal cancer. RNA molecules, often protected within extracellular vesicles or associated with proteins, can provide valuable insights into tumor activity and advances in other cancer types, such as breast cancer, where key RNA markers, such as *EPCAM*, *KRT19*, *ERBB2*, and others, have been identified. However, the full potential of circulating RNA in colorectal cancer remains underexplored [16].

This study aimed to address this gap by investigating the utility of circulating RNA transcripts in detecting colorectal cancer at different stages of progression, from benign adenomas to malignant carcinomas. By focusing on the ACS, which characterizes colorectal cancer development, this study aimed to identify novel circulating RNA biomarkers that could improve early detection; provide real-time insights into tumor biology; and contribute to the development of more sensitive, non-invasive screening tools. High-throughput RNA sequencing was employed to profile circulating RNA from healthy controls (HCs); symptomatic non-disease controls (NDCs); and patients with non-advanced adenomas, advanced adenomas, and colorectal cancer (Figure 1), with the goal of identifying differentially expressed genes (DEGs) that can enhance diagnostic accuracy and prognostication [4,17].

## 2. Results

### 2.1. The Selection of Circulating Transcripts Associated with the ACS via RNA Sequencing

To identify novel circulating transcripts associated with colorectal cancer progression, RNA sequencing was performed on 100 samples (20 from each group: HC, NDC, non-advanced adenoma, advanced adenoma, and colorectal cancer). Of the 46,427 initial genes, 14,844 were analyzed after excluding those with a count value of zero in at least one sample. DEGs between groups were identified using a |log_2_ fold change| of ≥2 and a *p* value of < 0.05. A total of 187 significant DEGs were detected across 10 comparison pairs (NDC vs. HC, advanced adenoma vs. non-advanced adenoma, and colorectal cancer vs. advanced adenoma). These DEGs were further examined for their biological significance to the adenoma–carcinoma sequence (Appendix A).

### 2.2. Circulating Transcripts in the Transition from HC to NDC

We identified 24 significant DEGs on comparison between the HC and NDC groups. The most prominent gene, *IFI27*, was also significantly upregulated. Functional annotation through a GO analysis showed that the identified genes were involved in RNA processing, specifically, mRNA splicing and immune-related processes. These findings suggest early immune system activation even in the absence of a confirmed pathology, as observed in the NDC group (Appendix A).

### 2.3. Transition from NDC to Non-Advanced Adenoma

Fourteen significant DEGs were identified in the non-advanced adenoma vs. NDC comparison. A GO analysis revealed that genes such as *DEFA4*, *IGHG1*, and *IGLC2* are involved in immune responses, especially antigen-binding and neutrophil-mediated defense mechanisms. This immune activation indicated that during the early stages of adenoma formation, the immune system is actively engaged in surveillance against tumor progression (Appendix A).

### 2.4. Transition from Non-Advanced Adenoma to Advanced Adenoma

Seventeen DEGs were observed during the transition from non-advanced adenoma to advanced adenoma, with genes such as *FCGR1A* and *S100P* showing reduced expression. These DEGs were associated with immune regulation and antibody-dependent cellular cytotoxicity. Interestingly, non-coding RNAs, such as *LOC107984755* and *RPL29P4*, were upregulated, reflecting a shift towards altered cellular regulation as the adenomas became more advanced (Appendix A).

### 2.5. Transition from Advanced Adenoma to Colorectal Cancer

A total of 86 DEGs were identified during the advanced adenoma-to-colorectal cancer transition. Notably, the upregulation of *CD177*, a marker of neutrophil activity, suggests a critical role of neutrophil-mediated responses in the transition from adenoma to carcinoma. Other key transcripts, such as *MPO* and *DEFA3*, further emphasize the role of neutrophil extracellular traps and their immune activity in tumor progression. The transition to malignancy is characterized by increased immune response activation and cellular damage repair mechanisms. The upregulation of immune-related genes such as *MPO* and *DEFA3* highlights their potential as biomarkers for the detection of advanced colorectal cancer stages (Appendix A).

### 2.6. Protein–Protein Interaction and Pathway Analyses

A STRING network analysis revealed strong interactions between key genes such as *MPO*, *DEFA4*, and *CD177* across the ACS. These interactions suggest that immune processes, particularly neutrophil-mediated responses, are central to colorectal cancer progression. A KEGG pathway analysis further supported this finding by identifying neutrophil extracellular trap formation and Fc gamma receptor-mediated phagocytosis as critical immune-related pathways in the adenoma–carcinoma transition (Figure 2, Figure 3, Figure 4, Figure 5 and Figure 6).

### 2.7. Clinical Validation of Candidate Biomarkers

An RT-qPCR analysis was performed to validate *FCGR1A* and *MPO* levels in 20 samples, each from the advanced adenoma, non-advanced adenoma, and HC groups. *FCGR1A* was significantly upregulated in the advanced adenoma group compared with the HC group, indicating its potential as an early biomarker of adenomas. Although upregulated in the non-advanced adenoma group, the *FCGR1A* difference was not statistically significant. *MPO*, however, was significantly upregulated in the non-advanced and advanced adenoma groups, confirming its role as a circulating biomarker throughout the ACS. These findings support the utility of *MPO* and *FCGR1A* as potential non-invasive biomarkers for colorectal cancer screening and monitoring, especially throughout the ACS (Figure 7).

## 3. Discussion

This study provides a comprehensive analysis of circulating RNA transcripts associated with colorectal cancer progression, particularly within the ACS. A total of 187 DEGs were identified with significant enrichment in immune response pathways, specifically, those involving neutrophil activity. Key transcripts, such as *MPO*, *FCGR1A*, *DEFA4*, and *CD177*, have been highlighted as potential biomarkers of colorectal cancer, underscoring the role of immune dysregulation in tumor development and progression [18,19].

Our findings emphasize the immune system’s pivotal role in CRC progression, particularly highlighting the involvement of neutrophils, which promote tumor development through mechanisms such as neutrophil extracellular traps (NETs) [20]. We found that DEFA4 is associated with non-advanced adenoma, MPO with advanced adenoma, and CD177 with CRC, suggesting their potential as biomarkers for distinct ACS stages. In the early stages of CRC, neutrophils, along with DEFA4, mediate an initial immune response to microbial and endogenous stimuli. As immune complexes, including cytokines, accumulate, FCGR1A, expressed on neutrophil membranes, interacts with these complexes, facilitating NETosis. During the advanced adenoma (AA) stage, NETosis leads to reactive oxygen species (ROS) production, triggering MPO release, which further amplifies cellular damage and promotes tumor progression [21]. Our study substantiates previous findings that neutrophils act as a ‘double-edged sword’, providing beneficial immune responses but, under certain conditions, exacerbating tumorigenesis. By elucidating the stage-specific roles of circulating RNA markers in neutrophil-driven immune mechanisms, our findings advance the understanding of CRC pathophysiology and highlight novel avenues for early detection and therapeutic intervention. The upregulation of *MPO* and *CD177*, markers closely associated with neutrophil activity, supports the hypothesis that innate immune responses are a driving force in the transition from adenomas to carcinoma [22,23].

Compared to prior studies focusing on tissue-based biomarkers or adaptive immune responses, our research, using circulating RNA from whole blood, provides a non-invasive alternative. The inclusion of immune-related transcripts, such as *MPO* and *FCGR1A*, in the early and advanced stages of colorectal cancer highlights the value of liquid biopsy for cancer detection [24]. Advanced adenomas (AAs) are associated with a 2.7-fold higher incidence of CRC and a 2.6-fold increase in CRC-related mortality compared to normal or non-advanced adenomas [25]. Therefore, early CRC detection at the advanced adenoma stage is critical for timely intervention and improved patient outcomes. This study highlights MPO and FCGR1A as particularly compelling biomarkers for both advanced adenoma detection and early CRC monitoring, addressing key clinical unmet needs in CRC diagnosis and prognosis. One of the significant challenges in clinical practice is the lack of clear guidelines on when to remove precancerous polyps and how to manage recurrent polyps, which places a considerable burden on clinicians. While the probability of a polyp progressing to malignancy is generally low, studies have reported an increased risk of metastasis triggered by immune responses following the surgical removal of recurrent polyps [26].

However, there is a lack of definitive evidence and reliable monitoring markers to guide these medical decisions. Our study provides novel insights into the role of neutrophils in CRC progression, particularly through MPO and FCGR1A, which may serve as critical markers in understanding immune-medicated tumor progression. A deeper characterization of neutrophil involvement, as revealed in this study, offers potential solutions to these unresolved clinical challenges and may contribute to refining CRC screening, surveillance, and treatment strategies. Although similar studies have explored the roles of ctDNA and CTCs, this study demonstrates the potential of RNA biomarkers, particularly those reflecting immune activity [27].

However, our study diverges from previous research that primarily emphasizes adaptive immune responses, such as T-cell involvement, in colorectal cancer progression [28]. In contrast, we found that innate immune responses, particularly those mediated by neutrophils, play a central role in the early stages of adenoma development. This difference could be attributed to the sample type (whole blood vs. tissue) and the broader representation of immune cell types, including neutrophils, in our analysis [29]. Additionally, whole-blood RNA sequencing provides a more holistic view of circulating immune cells and offers unique insights into colorectal cancer biology [30].

Despite these promising findings, this study has some limitations. The sample size for RNA sequencing was relatively small, and while the clinical validation of *MPO* and *FCGR1A* is encouraging, larger studies with more diverse patient populations are needed to confirm their diagnostic utility [31]. Furthermore, our study focuses on circulating RNA transcripts, which, when combined with other liquid biopsy markers, such as ctDNA and CTCs, could potentially yield even more robust diagnostic tools [32,33].

Future studies will be needed to validate these RNA biomarkers in larger multicenter cohorts and to explore their prognostic value over time. Additionally, integrating transcriptomic data with other circulating biomarkers could enhance diagnostic precision and provide insights into dynamic changes in tumor biology during colorectal cancer progression and treatment [34,35].

Overall, this study contributes to the growing body of evidence showing that neutrophil activity plays a significant role in colorectal cancer progression. The identification of circulating biomarkers such as *MPO* and *FCGR1A* could contribute to the development of non-invasive screening tools for early colorectal cancer detection, offering potential clinical benefits through more personalized approaches to colorectal cancer management.

## 4. Materials and Methods

### 4.1. Study Participants

This study was approved by the Institutional Review Boards of Severance Hospital (approval no. 4-2017-0148), Gangnam Severance Hospital (approval no. 3-2017-0024), Gangbuk Samsung Hospital (approval no. 2017-02-022-009), and the Medical Checkup Center of Wonju Severance Christian Hospital (approval no. CR319115). A total of 160 blood samples were collected from individuals scheduled for a colonoscopy at these institutions between 2017 and 2023, with all blood samples collected prior to the colonoscopy procedure. The participants included adults aged 19 years or older who provided written informed consent and were either scheduled for a colonoscopy during routine health screenings or who presented with gastrointestinal symptoms at a gastroenterology clinic. The exclusion criteria included a lack of consent, intellectual disabilities or severe psychiatric disorders, a history of malignancy or curative treatments within the previous 5 years, recent use of immunosuppressive drugs (within 6 months), and pregnancy. One hundred samples were used for next-generation sequencing and 60 samples for qualitative reverse transcription PCR (RT-qPCR).

Blood samples were divided into five groups based on the colonoscopy and histological results, including dysplasia grade level, villous component protein, size, and number of polyps, according to the European Society of Gastrointestinal Endoscopy. The samples were classified into colorectal cancer, advanced adenoma, non-advanced adenoma, NDC, and HC groups. The samples used in this study were randomly selected from each group and are summarized in Table 1.

### 4.2. Blood Collection and RNA Isolation

Blood samples (3 mL) were collected via venipuncture using Tempus^TM^ Blood RNA Tubes (Applied Biosystems, Chicago, IL, USA) to avoid epithelial cell contamination. The tubes were vortexed for 10 s to ensure mixing with a stabilizing reagent. Blood samples were stored at either 4 °C for up to seven days or frozen at −20 °C until used for RNA isolation. The total RNA was extracted using the Tempus^TM^ Spin RNA Isolation Kit (Applied Biosystems), following the manufacturer’s protocol. RNA quality was assessed using the Agilent 2200 TapeStation System (Agilent Technologies, Santa Clara, CA, USA). Samples with an RNA Integrity Number (RIN) greater than 7.0 were selected for further analysis.

### 4.3. cDNA Synthesis

Complementary DNA (cDNA) was synthesized using M-MLV reverse transcriptase (Invitrogen, Carlsbad, CA, USA); random hexamers (Invitrogen); and a dNTP mixture (Intron Biotechnology, Seongnam, Republic of Korea). cDNA was synthesized according to the manufacturer’s instructions.

### 4.4. RNA Sequencing and Differential Gene Expression Analysis

RNA sequencing was conducted at Macrogen (Seoul, Republic of Korea). The RNA concentration was quantified using the Quant-IT^TM^ RiboGreen RNA Assay Kit (Invitrogen, Carlsbad, CA, USA), and RNA Integrity was confirmed using the Agilent 2200 TapeStation System (Agilent Technologies). Samples with a RIN of >7.0 were selected for library construction. RNA libraries were prepared using the TruSeq^®^ Stranded Total RNA with Ribo-Zero Globin Kit (Illumina, San Diego, CA, USA) with poly (A) selection and fragmentation at 94 °C for 8 min, targeting an insert size of approximately 300 bp. Sequencing was performed at a depth of 30 million reads per sample to ensure sufficient coverage for the differential expression analysis.

### 4.5. Gene-Enrichment and Functional Annotation Analysis

A gene-enrichment analysis was performed using g:Profiler [RRID:SCR_006809] (Version e1 1o_eg57_p18_4b54a898) to evaluate the gene ontology (GO) and biological pathways. DEGs were selected using an adjusted *p* value of <0.05 and a log_2_ fold change (FC) of >2.

### 4.6. Clustering Heatmap Analysis

Clustering heatmaps for DEGs were generated using MultiExperiment Viewer (MeV Version 4.9.0), which is a Java-based desktop application for gene expression analysis and visualization.

### 4.7. Protein–Protein Interaction Network Analysis

Protein–protein interaction networks were constructed using STRING [RRID:SCR_005223] (Version 12.0), with an interaction score of >0.4 for medium confidence. Network topology parameters were calculated and visualized to explore the interaction clusters.

### 4.8. GO Analysis I

A GO analysis of DEGs was conducted using the ClueGO (RRID:SCR_005748) plugin (version 2.5.10) and the CluePedia plugin (Version 1.5.10) in Cytoscape [RRID:SCR_003032] (Version 3.10.0). Functional correlations were explored using hypergeometric testing, and significant pathways were selected for further investigation.

### 4.9. GO Analysis II

Gene Set Enrichment Analysis (GSEA) [RRID:SCR_003199] was used to assess concordant differences in predefined gene sets between the biological states. The enrichment scores (ESs) and *p* values were calculated based on the expression profiles.

### 4.10. Gene Pathway Analysis

Gene pathways were analyzed using the Kyoto Encyclopedia of Genes and Genomes (KEGG) [RRID:SCR_012773] database to gain insights into the biological functions and pathways involved in the sample groups.

### 4.11. Quantitative PCR Assay

Quantitative PCR (qPCR) was performed to quantify gene expression using a StepOnePlus^TM^ Real-Time PCR System software v2.0.2 (Applied Biosystems). The quantification cycle (Cq) method was used, with *GAPDH* as the reference gene for normalization. Thermal cycling conditions were set at 95 °C for 10 min, followed by 40 cycles at 95 °C for 15 s and at 60 °C for 1 min. Relative gene expression levels were calculated using the 2^−ΔCq^ method. All qPCR reactions included non-template controls and were performed in triplicate for each sample to ensure reproducibility.

### 4.12. Statistical Analysis

All statistical analyses were performed using GraphPad Prism [RRID:SCR_002798] (version 9.0). Differences between groups were assessed using Student’s *t*-test. Statistical significance was set at *p* < 0.05.

## 5. Conclusions

This study identified key circulating RNA transcripts, including *MPO*, *FCGR1A*, *DEFA4*, and *CD177*, that play crucial roles in ACS progression. These results highlight the importance of immune responses, particularly neutrophil-mediated mechanisms, in colorectal cancer development. By focusing on circulating RNA biomarkers, this study offers a promising non-invasive approach for detecting and monitoring colorectal cancer. Clinical validation of *MPO* and *FCGR1A* underscores their potential as biomarkers for early adenoma detection and advanced colorectal cancer monitoring. Future research should validate these findings in larger cohorts and integrate them with other biomarkers to enhance colorectal cancer screening accuracy, potentially leading to earlier interventions and more personalized treatment strategies.

## Figures and Tables

**Figure 1 ijms-26-01518-f001:**
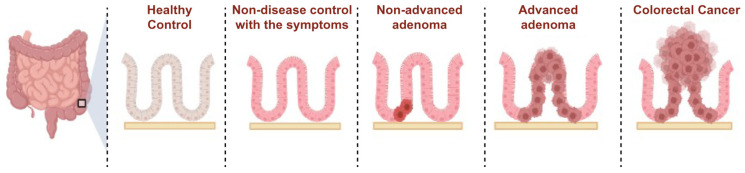
The adenoma–carcinoma sequence in colorectal cancer progression. The adenoma–carcinoma sequence is a model that represents colorectal cancer progression and is a fundamental concept in colorectal cancer research. Colorectal cancer progresses from normal epithelial cells to non-diseased controls, non-advanced adenoma, advanced adenoma, and cancer. Advanced adenomas, which are in the precancerous stage, have high-grade dysplastic lesions. An advanced adenoma is defined when it contains more than 25% of cells in the villous form, larger than 10 mm, or has three or more polyps. In advanced adenomas, the incidence and mortality rates of colorectal cancer are more than double those of normal and non-advanced adenomas. Figures were created using BioRender.com (accessed on 20 August 2023).

**Figure 2 ijms-26-01518-f002:**
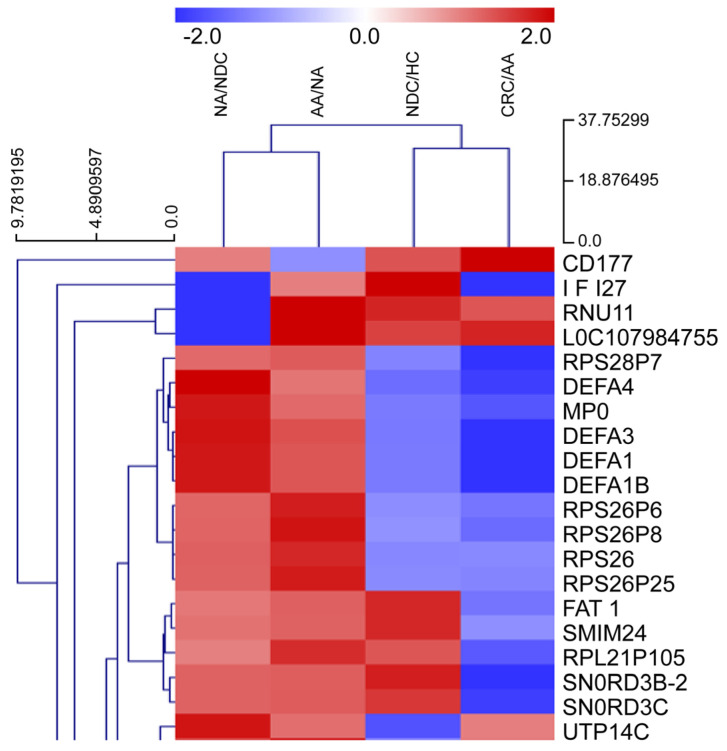
Clustering heatmap analysis. The fold change patterns of the top 13 genes among 187 DEGs in the adenoma–carcinoma sequence (ACS). This heatmap allows for a direct comparison of the five-stage change patterns from the HC group to the CRC group in the ACS. Among the top 13 hierarchical clustering transcripts, *CD177* is the most expressed transcript during the AA group to CRC group transition (log_2_ FC: 6.75; *p* value: 3.1 × 10^−8^. Descriptions of the top 13 genes are shown in Appendix A. Red: increased fold change, Blue: decreased fold change.

**Figure 3 ijms-26-01518-f003:**
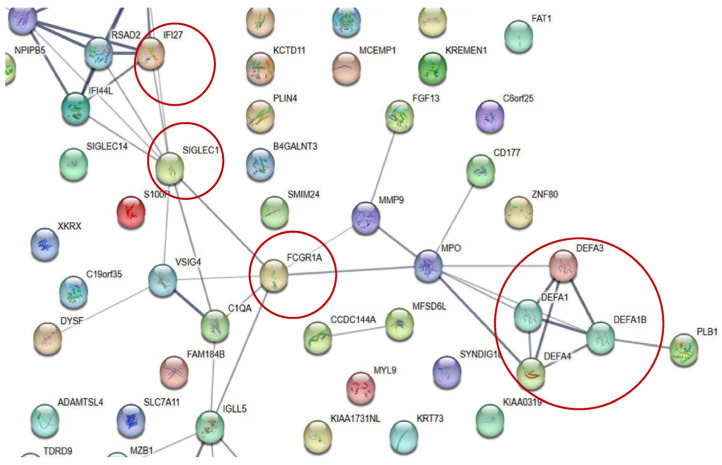
Protein–protein interactions. *IFI27*, *SIGLEC1, FCGR1A,* and *MPO* have connections to *DEFAs* and *CD177,* respectively. The protein–protein interaction scores are shown in Appendix A. The red circles mean that the networks were able to visually distinguish between the differentially expressed genes(DEGs) related to each stage. *IFI27*, which showed the highest fold change in the NDG group compared to that in the HC group, was linked to *SIGLEC1*, which was linked to *FCGR1A* and *MPO. MPO* was associated with *DEFA4, DEFA3, DEFA1*, and *DEFA1B,* which were highly expressed in the NA group, and with *CD177*, A representative gene in the CRC group. These results suggest that the mechanism is a sequential reaction that begins with *IFI27* and ends with *CD177*.

**Figure 4 ijms-26-01518-f004:**
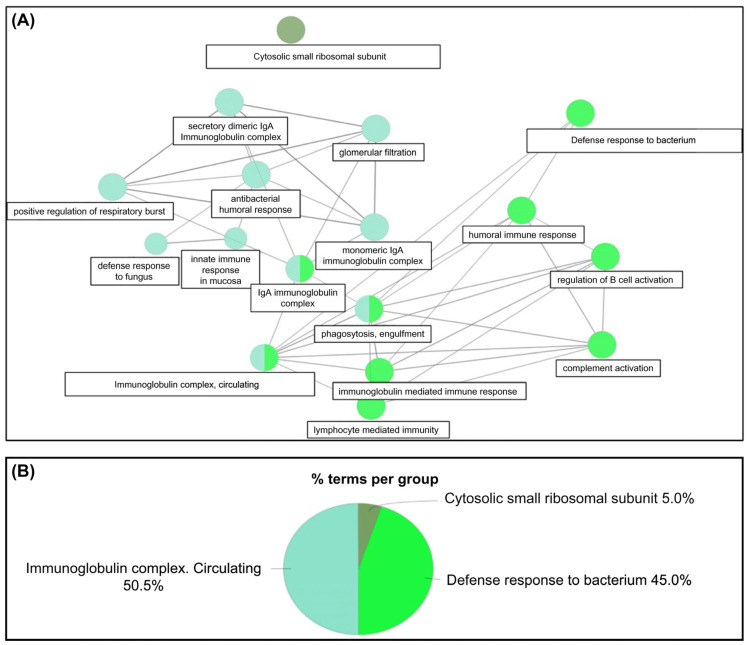
Gene ontology of 187 DEGs during the ACS of CRC in ClueGO. (**A**) The 187 circulating transcripts expressed during the ACS of colorectal cancer are genes corresponding to a circulating immunoglobulin complex, a defense response to a bacterium, and a cytosolic small ribosomal subunit. (**B**) The gene ontology terms are circulating immunoglobulin complex (50%), defense response to a bacterium (45%), and cytosolic small ribosomal subunit (5%). DEGs: Differentially expressed genes; ACS: Adenoma–carcinoma sequence; CRC: Colorectal cancer. The associated gene ontology terms are shown in Appendix A.

**Figure 5 ijms-26-01518-f005:**
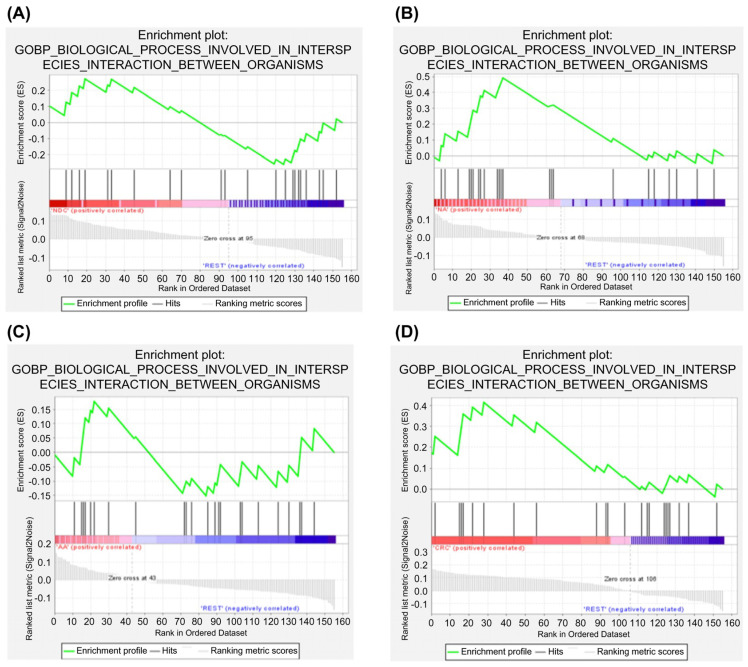
Gene enrichment analysis of biological processes involved in interspecies interactions between organisms. (**A**) When comparing the DEGs of the NDC group with those of the remaining groups (HC, NA, AA, and CRC), *IFI27* was shown to be the most important and significant circulating RNA. (**B**) When comparing the DEGs of the NA group with those of the remaining groups (HC, NDC, AA, and CRC), *DEFA4* was shown to be the most important and significant circulating RNA. (**C**) When comparing the DEGs of the AA group with those of the remaining groups (HC, NDC, NA, and CRC), *MPO* was shown to be the most important and significant circulating RNA. (**D**) When comparing the DEGs of the CRC group with those of the remaining groups (HC, NDC, NA, and AA), *CD177* was shown to be the most important and significant circulating RNA. The genes correlated with GSEA analysis in the comparison group are shown in Appendix A.

**Figure 6 ijms-26-01518-f006:**
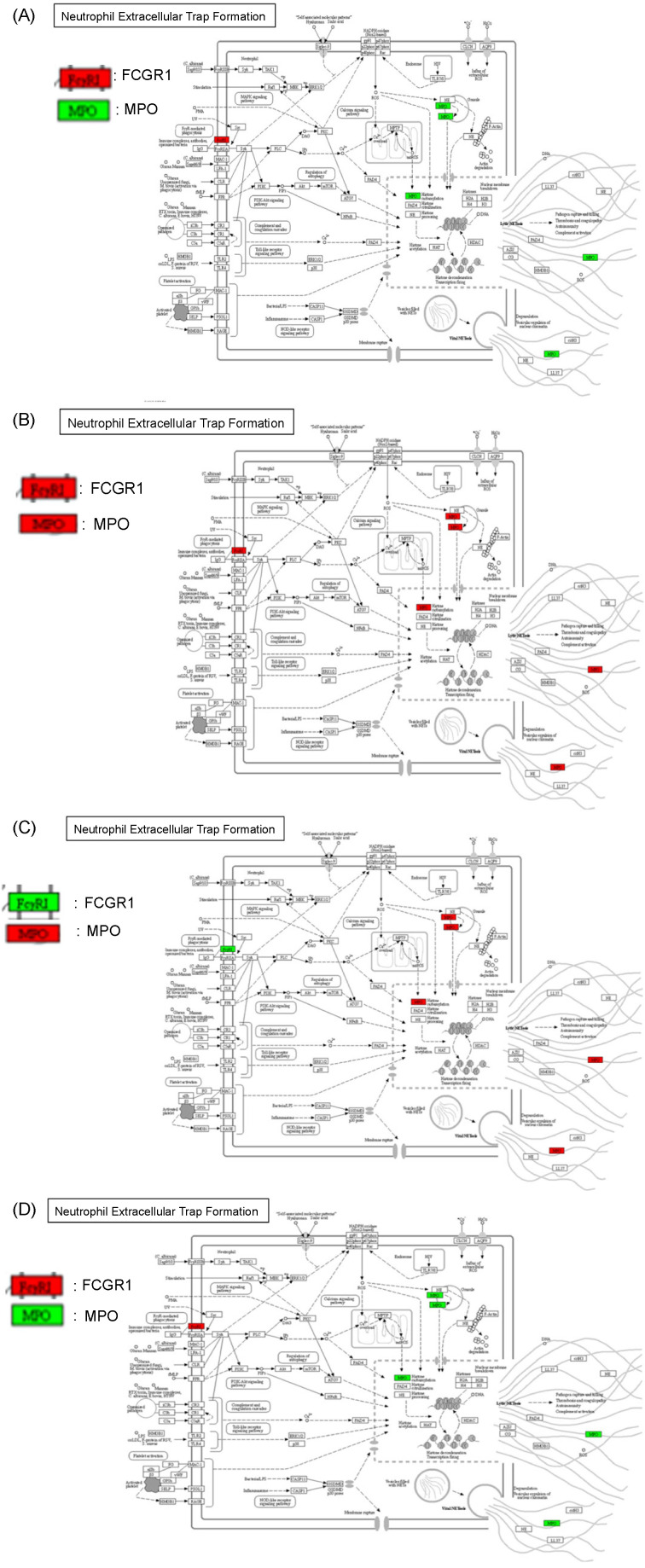
Fold change pattern of proteins in the neutrophil extracellular trap formation with KEGG pathway map viewer. (**A**) The fold change pattern of circulating transcripts reflects the transition of proteins. The transition pattern of *FCGR1* and *MPO* during the transition from the HC group to the NDC group in the neutrophil extracellular trap formation pathway. Compared to the HC group, the NDC group showed a pattern in which *FCGR1* in the cell membrane of neutrophils was upregulated, and *MPO* in the neutrophil granule was down-regulated. Red: *FCGR1*; green: *MPO*. (**B**) The activation pattern of *FCGR1* and *MPO* during the transition from the NDC group to the NA group in the neutrophil extracellular trap formation pathway. Compared to the NDC group, the NA group showed a pattern in which *FCGR1* in the cell membrane of neutrophils was upregulated, and *MPO* in the neutrophil granule was upregulated. Red: *FCGR1*; red: *MPO*. (**C**) The activation pattern of *FCGR1* and *MPO* during the transition from the NA group to the AA group in the neutrophil extracellular trap formation pathway. Compared to the NA group, the AA group showed a pattern in which *FCGR1* in the cell membrane of neutrophils was down-regulated, and *MPO* in the neutrophil granule was upregulated. Green: *FCGR1*; red: *MPO*. (**D**) The activation pattern of *FCGR1A* and *MPO* converting from the AA group to the CRC group in the neutrophil extracellular trap formation pathway. Compared to the AA group, the CRC group showed a pattern in which *FCGR1* in the cell membrane of neutrophils was upregulated, and *MPO* in the neutrophil granule was down-regulated. Red: *FCGR1*; green: *MPO*.

**Figure 7 ijms-26-01518-f007:**
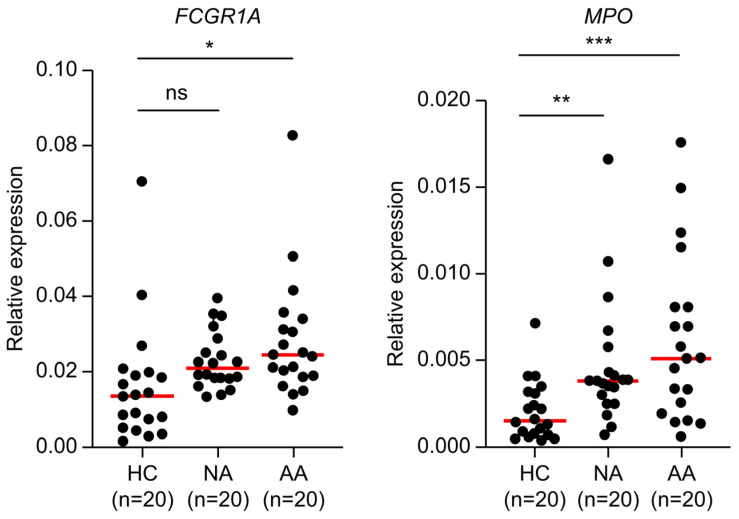
The results of RT-qPCR for *FCGR1A* and *MPO* expression of clinical samples. *MPO* was upregulated in both the NA group and AA group compared to the HC group, and they were statistically significant. *FCGR* was slightly upregulated in the NA group compared to the HC group, but there was no statistical significance. In the AA group, *FCGR1A* was upregulated and had statistical significance compared to the HC group. *p* (*, **, and ***, indicate *p* < 0.05, *p* < 0.01, and *p* < 0.001, respectively, ns means no statistically significance) was calculated by a two-tailed Student’s test. The information on the primers and probes for *FCGR1* and *MPO* are shown in Appendix A. Red line: Mean of relative expression.

**Table 1 ijms-26-01518-t001:** Criteria for group classification of colorectal cancer and the number of samples in each group.

Group	Description	No. of Samples	Collection Site
Colorectal cancer	Tissue damage and invasion; Cancer in colorectum;	20	(a), (b), and (c)
AA	Dysplasia: high grade. Villous component: more than 25%; Size: larger than 10 mm; Number: more than 3 in colorectum;	40	(a), (b), and (c)
NA	Dysplasia: low grade. Villous component: less than 25%; Size: smaller than 10 mm; Number: less than 3 in colorectum.	40	(a), (b), and (c)
NDC	No pathologic histological characteristics with symptoms such as stomach pain or abnormal bowel trouble.	20	(a), (b), and (c)
HC	No symptom; No colorectal disease.	40	(a), (b), (c), and (d)

AA: Advanced adenoma; NA: Non-advanced adenoma; NDC: Non-disease control with symptoms; HC: Healthy control. (a) Severance Hospital, Seoul, Republic of Korea; (b) Gangnam Severance Hospital, Seoul, Republic of Korea; (c) Gangbuk Samsung Hospital, Seoul, Republic of Korea; (d) Wonju Severance Christian Hospital, Wonju, Republic of Korea.

## Data Availability

The original contributions presented in this study are included in the article/Appendix A. Further inquiries can be directed to the corresponding author.

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
