# Peer review of "Circulating RNA Markers Associated with Adenoma–Carcinoma Sequence in Colorectal Cancer"

_ijms, 2025, doi:10.3390/ijms26041518_

Round 1
Reviewer 1 Report
Comments and Suggestions for Authors
The study investigated the early detection of the colorectal tumor through the use of liquid biopsy and the circulating RNA transcripts. The study was original and the findings could be useful in the clinical practice. The limits of the research are: the small size of study population and the use of only RNA transcript, without explorating ct DNA and circulating tumor cells.
The results are really interesting for the potential clinical and therapeutic use.
Author Response
Reviewer 1.
Comment: The study investigated the early detection of the colorectal tumor through the use of liquid biopsy and the circulating RNA transcripts. The study was original and the findings could be useful in the clinical practice. The limits of the research are: the small size of study population and the use of only RNA transcript, without explorating ct DNA and circulating tumor cells.
The results are really interesting for the potential clinical and therapeutic use.
Response to Reviewer 1:
Thank you for your positive evaluation of our study and your acknowledgment of its originality and potential clinical significance. We deeply appreciate your insightful comments on the limitations of our research, particularly regarding the small study population and the exclusive focus on RNA transcripts without exploring circulating tumor DNA (ctDNA) and circulating tumor cells (CTCs).
We acknowledge that the sample size is relatively small, which may limit the generalizability of our findings. We have clarified this limitation and future validation plans in the Discussion section. (3. Discussion, lines 269-276)
Regarding the exclusive use of RNA transcripts, we agree that incorporating other biomarkers such as ctDNA and CTCs would provide a more comprehensive understanding of the molecular mechanisms underlying colorectal cancer. We intend to integrate these additional biomarkers in future research to strengthen our findings and expand the scope of our study.
Once again, we thank you for your valuable feedback, which has been instrumental in refining our manuscript and guiding our future research directions.
Reviewer 2 Report
Comments and Suggestions for Authors
This paper is well done, but some questions must be explained in the final version.
What distinguishes this study from the previous ones in detecting the adenoma-carcinoma sequence (ACS) using circulating RNA biomarkers?
The authors state, 'Our findings align with previous research demonstrating the critical involvement of immune cells in colorectal cancer, particularly neutrophils, which are known to promote tumor progression through mechanisms such as neutrophil extracellular traps.' What is novel about these results, specifically regarding the roles of the identified circulating RNA transcripts (MPO, FCGR1A, DEFA4, and CD177) in elucidating neutrophil-mediated immune mechanisms in colorectal cancer?
What makes MPO and FCGR1A particularly compelling as biomarkers for early adenoma detection and advanced colorectal cancer monitoring? Please elaborate on these points in the final text.
The authors state in conclusion, "Future studies should aim to validate these RNA biomarkers in larger multicenter cohorts and explore their prognostic value over time." So, is this data correct, or is it a proposition?
Author Response
Reviewer 2.
This paper is well done, but some questions must be explained in the final version.
Comment 1: What distinguishes this study from the previous ones in detecting the adenoma-carcinoma sequence (ACS) using circulating RNA biomarkers?
Response to comment 1: This study is distinct in that it investigates circulating RNA biomarkers in human blood for detecting the adenoma-carcinoma sequence (ACS) in colorectal cancer (CRC). While previous research has explored circulating tumor DNA (ctDNA) and extracellular miRNAs for CRC detection, these approaches primarily reflect genetic mutations (ctDNA) or passive RNA release (miRNAs) rather than active transcriptional changes occurring within tumor-associated cells.
In contrast, our study uniquely identifies intracellular circulating RNA transcripts (e.g., MPO, FCGR1A, DEFA4, and CD177) that are functionally linked to the host immune response, particularly neutrophil-mediated mechanisms. These biomarkers provide a real-time snapshot of immune dynamics during ACS progression, distinguishing them from ctDNA-based genetic alterations or extracellular miRNAs, which do not necessarily indicate active cellular responses or immune interactions.
Moreover, previous studies on blood-based ACS detection have largely relied on methylated DNA markers (e.g., mSEPT9), which exhibit high specificity for advanced CRC but limited sensitivity for early adenomas [1]. Similarly, CTC-based assays focus on epithelial-to-mesenchymal transition (EMT) markers, which are more relevant for late-stage CRC but do not fully capture immune-mediated processes in early disease stages [2].
By focusing on intracellular circulating RNA transcripts, our study provides a functional and dynamic framework for ACS detection, bridging a critical gap in understanding early immune responses and tumor-microenvironment interactions. To emphasize this distinction, we have revised the abstract accordingly (Abstract, lines 19–22, 28–30 and introduction, line 61-72).
[12] Shen, L.; Fang, J.; Qiu, Z.; Zhao, Y.; Zou, H. Methylated SEPT9 (mSEPT9): A Promising Blood-Based Biomarker for Colorectal Cancer Detection. Int. J. Mol. Sci. 2022, 23, 4567.
[13] Pantel, K.; Alix-Panabières, C. Circulating Tumor Cells for Early Detection of Clinically Relevant Cancer. Nat. Rev. Clin. Oncol. 2023, 20, 453–467.
[14] Choi, C.H.; Kim, Y.T.; Song, S.Y.; Kim, C.G.; Lee, J.H.; Park, S.W. MicroRNA Expression Signatures in Colorectal Adenoma and Carcinoma Progression. Clin. Epigenetics 2016, 8, 30.
[15] Kanaan, Z.; Rai, S.N.; Eichenberger, M.R.; Roberts, H.; Keskey, R.; Pan, J.; Yan, Y.; Jorden, J.; Denning, W.L.; Korkaya, H.; et al. Complex Patterns of Altered MicroRNA Expression During the Adenoma-Carcinoma Sequence in Colorectal Cancer. Clin. Cancer Res. 2011, 17, 7283–7293
Comment 2: The authors state, 'Our findings align with previous research demonstrating the critical involvement of immune cells in colorectal cancer, particularly neutrophils, which are known to promote tumor progression through mechanisms such as neutrophil extracellular traps.' What is novel about these results, specifically regarding the roles of the identified circulating RNA transcripts (MPO, FCGR1A, DEFA4, and CD177) in elucidating neutrophil-mediated immune mechanisms in colorectal cancer?
Response to comment 2: Our findings emphasize the immune system’s pivotal role in colorectal cancer (CRC) progression, particularly highlighting the involvement of neutrophils, which promote tumor development through mechanisms such as neutrophil extracellular traps (NETs).
We found that DEFA4 is associated with non-advanced adenoma, MPO with advanced adenoma, and CD177 with colorectal cancer, suggesting their potential as biomarkers for distinct ACS stages. In the early stages of CRC, neutrophils, along with DEFA4, mediate an initial immune response to microbial and endogenous stimuli. As immune complexes, including cytokines, accumulate, FCGR1A, expressed on neutrophil membranes, interacts with these complexes, facilitating NETosis. During the advanced adenoma (AA) stage, NETosis leads to reactive oxygen species (ROS) production, triggering MPO release, which further amplifies cellular damage and promotes tumor progression.
Our study substantiates previous findings that neutrophils act as a double-edged sword—providing beneficial immune responses but, under certain conditions, exacerbating tumorigenesis. By elucidating the stage-specific roles of circulating RNA markers in neutrophil-driven immune mechanisms, our findings advance the understanding of CRC pathophysiology and highlight novel avenues for early detection and therapeutic intervention.
To emphasize our study’s novelty, we have revised the Discussion section, incorporating key points in lines 223-238.
Comment 3: What makes MPO and FCGR1A particularly compelling as biomarkers for early adenoma detection and advanced colorectal cancer monitoring? Please elaborate on these points in the final text.
Response to comment 3:
As the reviewer’s comment, we have provided a detailed explanation of why MPO and FCGR1A are compelling biomarkers for advanced adenoma (AA) detection and early colorectal cancer (CRC) monitoring, emphasizing their role in addressing key clinical unmet needs in CRC diagnosis and prognosis. To clarify their significance, we have revised the discussion section accordingly and included additional supporting details. The updated content can be found in 3. Discussion, lines 246–264."
Advanced adenomas(AA) are associated with a 2.7-fold higher incidence of colorectal cancer (CRC) and a 2.6-fold increase in CRC-related mortality compared to normal or non-advanced adenomas [25]. Therefore, early CRC detection at the advanced adenoma(AA) stage is critical for timely intervention and improved patient outcomes. This study highlights MPO and FCGR1A as particularly compelling biomarkers for both advanced adenoma(AA) detection and early CRC monitoring, addressing key clinical unmet needs in CRC diagnosis and prognosis. One of the significant challenges in clinical practice is the lack of clear guidelines on when to remove precancerous polyps and how to manage recurrent polyps, which places a considerable burden on clinicians. While the probability of a polyp progressing to malignancy is generally low, studies have reported an increased risk of metastasis triggered by immune responses following surgical removal of recurrent polyps.
However, there is a lack of definitive evidence and reliable monitoring markers to guide these medical decisions. Our study provides novel insights into the role of neutrophils in CRC progression, particularly through MPO and FCGR1A, which may serve as critical markers in understanding immune-mediated tumor progression. A deeper characterization of neutrophil involvement, as revealed in this study, offers potential solutions to these unresolved clinical challenges and may contribute to refining CRC screening, surveillance, and treatment strategies.
[25] Click B.; Pinsky PF.; Hickey T.; Doroudi M.; Schoen RE. Association of Colonoscopy Adenoma Findings With Long-term Colorectal Cancer Incidence. JAMA 2018 May 15;319(19):2021-2031.
Comment 4: The authors state in conclusion, "Future studies should aim to validate these RNA biomarkers in larger multicenter cohorts and explore their prognostic value over time." So, is this data correct, or is it a proposition?
Response to comment 4: Thank you for your question. The statement in the conclusion, "Future studies should aim to validate these RNA biomarkers in larger multicenter cohorts and explore their prognostic value over time," is a proposition, not a statement based on existing data. It reflects our future research direction to enhance the clinical applicability of our findings. To clarify this point, we have slightly revised the conclusion to explicitly indicate that this is a proposed direction for future research. (Discussion, lines 281-282)
Round 2
Reviewer 2 Report
Comments and Suggestions for Authors
After all the changes have been made, the paper is ready to go